# Prenatal Workshops and Support Groups for Prospective Parents Whose Children Will Need Neonatal Care at Birth: A Feasibility and Pilot Study

**DOI:** 10.3390/children10091570

**Published:** 2023-09-19

**Authors:** Béatrice Boutillier, Guillaume Ethier, Isabelle Boucoiran, Martin Reichherzer, Thuy Mai Luu, Lucie Morin, Rebecca Pearce, Annie Janvier

**Affiliations:** 1CHU Sainte-Justine Research Center, Montréal, QC H3T 1C5, Canada; beatrice.boutillier@umontreal.ca (B.B.); isabelle.boucoiran@umontreal.ca (I.B.); thuy.mai.luu@umontreal.ca (T.M.L.); lucie.morin.2@umontreal.ca (L.M.); 2Division of Neonatology, CHU Sainte-Justine, Montréal, QC H3T 1C5, Canada; guillaume.ethier.hsj@ssss.gouv.qc.ca (G.E.); martin.reichherzer.hsj@ssss.gouv.qc.ca (M.R.); 3Unité D’éthique Clinique, CHU Sainte-Justine, Montréal, QC H3T 1C5, Canada; 4Division of Maternal-Fetal Medicine, CHU Sainte-Justine, Montréal, QC H3T 1C5, Canada; 5Department of Obstetrics and Gynecology, Université de Montréal, Montréal, QC H3T 1J4, Canada; 6Department of Pediatrics, Université de Montréal, Montréal, QC H3T 1C5, Canada; 7Parent Representative, Collaborates with Canadian Premature Babies Foundation, Etobicoke, ON M8X 1Y3, Canada; rebecca.pearce@gmail.com; 8Bureau de L’éthique Clinique (BEC), Université de Montréal, Montréal, QC H3C 3J7, Canada; 9Unité de Soins Palliatifs, CHU Sainte-Justine, Montréal, QC H3T 1C5, Canada

**Keywords:** peer-to-peer support, prenatal workshop, prenatal classes, high-risk pregnancy, prematurity, intensive care unit, parental perspectives, family-centered care, family integrated care, family partnership, patient partnership, parent partnership, pregnancy, neonatal care, neonatal intensive care unit

## Abstract

**Introduction**: Support groups in neonatal intensive care units (NICUs) are beneficial to parents. The usefulness of prenatal support groups for prospective parents who will have a newborn requiring admission to the NICU has never been investigated. **Methods**: We assessed the needs of NICU parents regarding topics they would have wished to discuss prenatally and developed the content of a prenatal support workshop. A standardized survey prospectively evaluated the perspectives of pregnant women admitted to a high-risk pregnancy unit who participated in the resulting workshops. **Results**: During needs assessment, 295 parents invoked themes they would have wished to discuss antenatally: parental guilt, future parental role, normalizing their experience/emotions, coping with many losses, adapting to their new reality, control and trust, information about the NICU, technology around the baby, common neonatal interventions, the NICU clinical team, and the role of parents in the team. These findings were used to develop the workshop, including a moderator checklist and a visual presentation. Practical aspects of the meetings were tested/finalized during a pre-pilot phase. Among 21 pregnant women who answered the survey (average gestational age 29.3 weeks), all agreed that the workshop was useful, that it made them feel less lonely (95%), that exchanges with other women were beneficial (95%) and gave them a certain amount of control over their situation (89%). All answers to open-ended questions were positive. **Conclusion**: Prenatal educational/support workshops provide a unique and useful means to support future NICU parents. Future investigations will explore whether these prenatal interventions improve clinical outcomes.

## 1. Introduction

The philosophy of care in neonatal intensive care units (NICUs) has undergone major transformations in the last decades. In the past, parents had limited involvement in the care of their babies.

Investigations have demonstrated that the outcome of children, parents, and families were improved with increased parental involvement. Patient-centered care became the norm, and eventually gave place to family-centered care and family integrated care. Parents’ participation in their infant’s care during hospitalization is now encouraged and positive impacts on the baby’s health are recognized [1,2]. Parents are now part of the care team. Despite these changes, being a parent in the NICU continues to have challenges [3,4]. Parents are often mourning a “normal” pregnancy, delivery, and/or baby, and may experience guilt, anxiety, and sadness. Furthermore, because of the medical technologies needed, bonding and parenting are more complex.

For over a decade, our PAF team (Partenariat Famille: family partnership committee) has performed several quality improvement interventions to improve the experience of parents and families in our unit [5,6,7]. After a feasibility study, we recently developed workshops for parents in the NICU [8,9]. One of these workshops is called “being a parent in the NICU” and has been evaluated as beneficial by families [9]. The following were the main themes reported by parents when explaining why these meetings helped: decreasing isolation and becoming a community, getting practical information, and promoting hope and resilience. Through sharing stories with other parents who also had experienced loss, sadness, and grief, parents trusted that it was possible to adapt and thrive. The meetings normalized parents’ emotions, decreased negative emotions (e.g., anger, sadness, guilt), empowered them in their parental role (and their added value in the clinical team caring for their child), and helped them communicate with loved ones and providers.

Prospective parents in “normal” pregnancies have the opportunity to engage in prenatal classes as a group, but these classes or groups did not exist for parents expecting a sick baby requiring NICU care (nor for sick pregnant women with pathological pregnancies), nor did they address the potential needs of future “NICU parents.” There is also no literature on how to develop such prenatal workshops nor the content that prospective parents may find useful. Prenatal classes and groups have educational aims (describing the pregnancy and delivery process) but also include a group experience where parents can engage with others in their situation.

This study had two objectives. The first objective was to examine whether parents in the NICU and/or after an NICU experience would find a prenatal workshop beneficial and what topics they would wish to have discussed. The second objective was to create the workshop and examine whether it was useful to prospective parents whose babies would likely be admitted to the NICU at birth.

## 2. Materials and Methods

The Neonatal PAF (“Partenariat Famille”) team was created in 2011 to recruit and coordinate family stakeholders and optimize the care of families in the NICU at Sainte- Justine University Health Center (CHUSJ), a mother–child quaternary care university-affiliated hospital in Montreal, Canada. The NICU is a 67-bed unit (single patient rooms) with about 1000 admissions per year (two-thirds of which are inborn patients).

The following two steps were followed to develop and implement prenatal workshops for prospective parents whose baby would be admitted to the NICU. Firstly, the development and implementation of workshops based on the results of a needs assessment and an investigation performed among former NICU parents. Secondly, a prospective evaluation of the impact of the workshops was performed.

### 2.1. Needs Assessment

The needs assessment phase examined whether prenatal workshops would be of interest to families and if so, which topics parents would want to discuss. Two studies helped us answer those questions.

In 2018, during a longitudinal evaluation of bi-weekly postnatal peer-support meetings in a tertiary care NICU, participating parents were asked to evaluate meetings on a scale of 0 to 10; with open-ended questions, they were asked about their perspectives. The results of this study have been published [5]. In one open-ended question, we also asked parents how we could better support them; these results have not been previously reported.

The second study was the Parents’ Voices Project, a Canadian initiative aimed at redefining outcomes that are important to parents in neonatology. It involved a prospective questionnaire of parents of extremely preterm children (˂29 weeks’ gestational age) seen at the neonatal follow-up clinic of the CHUSJ. For one year (July 2018–July 2019), all parents of children who were scheduled for a neonatal follow-up clinic visit were approached at the time of the appointment to participate.. A questionnaire was developed with parent partners and the PAF team. Through closed and open-ended questions, parents were asked about their perspectives, including what they wish they had known before birth.

Based on those results, the PAF group made a list of themes that parent participants reported they wished to hear antenatally. We then developed a visual slide presentation (including pictures) and a moderator checklist. Finally, we planned weekly workshops for pregnant women hospitalized in the high-risk pregnancy unit; their partner and families were also invited to participated. In 2020, because of COVID-19, the pre-pilot and the pilot phases were initiated in a virtual format, leading to other reported parental needs for online meetings, hybrid and in-person meetings.

### 2.2. Pilot Phase

In 2020, the “Ombrelles” (English translation: Umbrella) program was developed in the high-risk pregnancy unit. This mental health promotion program was designed for pregnant women admitted in the unit. It includes a number of workshops that address delivery plans, pain control during labor, and other topics. [10,11]. The PAF prenatal workshop, named “welcome to neonatology,” was introduced as part of the “Ombrelles” program, and was prospectively evaluated as a pilot study. Each workshop was moderated by a clinician member of the PAF team (nurse, fellow, neonatologist or neonatal nurse practitioner). During the week following each workshop, participating pregnant women were invited to give their feedback using a RedCap online questionnaire (interviews were not permitted because of COVID-19). Women who attended more than one meeting were only surveyed once. This questionnaire comprised of closed questions with responses on a 5-point Likert scale, as well as open-ended questions, which asked participants to evaluate the meeting, what they appreciated about it, what they disliked, and what they thought could be improved.

### 2.3. Data Analysis

Data are reported as descriptive statistics. Open-ended questions were analyzed using thematic analysis. Themes and sub-themes were rigorously defined by two independent investigators. Then, two independent investigators independently coded all the data. A specialist in mixed methods supervised each step of the analysis and assessed coding reliability. Discrepancies were resolved through consensus.

### 2.4. Ethics

All participants gave consent to participate. These projects (evaluation of pre- and postnatal workshops, Parents’ Voice Project, and the evaluation of the “Ombrelles” program) were approved by the Research Ethics Committee of the CHU Sainte-Justine Research Center.

## 3. Results

### 3.1. Needs Assessment

#### 3.1.1. Recommendations of Parents Who Participated in Postnatal Workshops

In total, 45 parents participated in the evaluation of the postnatal workshops [9]: Of these participants, 14 were fathers and 34 were mothers whose babies were mostly premature infants. In their responses to the open-ended questions, 32 (71%) mentioned that they would have benefited from and/or recommended a similar workshop to happen before birth. The topics parents wished to discuss gravitated around three themes: (1) Validating the parent’s experience and emotions and decreasing parental isolation; (2) the parental role in the NICU: when a parent could be present at the bedside, what a parent could do, what a baby would look like, and what they could do as parents to help their baby; and (3) educational aspects of the NICU (how the NICU works): clinicians taking care of the baby, technology around the baby, length of stay and so on.

The following quotes illustrate parental recommendations:

“*This workshop has helped me feel less isolated and that what I feel is normal. I shared my story with parents in my situation. I have become friendly with other mothers in the unit because of this. We support each other now. This kind of activity could happen before birth, where I felt very lonely and isolated on the prenatal floor*”(mother of a baby born at 25 weeks);

“*It is so much time to get to understand what a parent does in the NICU, what my role is as a father, what I can do, how I can be part of the team. This workshop has helped me so much, but it would have been even better if I had heard all this before birth. We knew we were coming here and there are no prenatal classes for parents that have a very abnormal pregnancy and will have an abnormal birth and a sick baby. My friends with normal pregnant wives have a group to prepare their births and I felt no control over our situation*”(Father of a baby born at 29 weeks);

“*Hearing that my feelings are normal, it is such a relief. Knowing that it is normal to not feel like a whole parent helps. Understanding how the intensive care works is important and this workshop helped a lot. It would have helped me even more before birth. I mean, I need to deconstruct so much guilt, anger, at the same time as understanding who takes care of my baby and how everything works in this new place, getting organized. It is not easy. There were really months to get prepared in my situation. We knew my daughter would be admitted here four months before she was born*” (Mother of a child with a congenital anomaly).

#### 3.1.2. Recommendations from Parents (The Parents’ Voice Study)

In the Parents’ Voice study, 248 parents of 213 children (83% of eligible children) provided 285 individual responses (some parents answered several questionnaires as their children were twins or triplets). Both parents answered the questionnaire for 71 children; there were only maternal answers for 128 children and only paternal answers for 14. The 213 children were born at a mean gestational age of 26.6 weeks with a mean birth weight of 907 g. A detailed description of participants can be found in a previous publication [12]. When asked in an open-ended question what they wished clinicians would have told them, 50% of parents (124) reported there was nothing more they wished they had learned/heard. Of the remaining 124 parents who wished they had learned/received more information, 67 invoked information they would have wished to receive prenatally. These prenatal themes reflected three main themes:(1)The reason for preterm birth, why their infant was born preterm and what they could have done, often associated with a feeling that they could have prevented the preterm birth.(2)Parent-centered topics: Being a parent in the NICU, what parents can do, the emotional burden of the hospitalization, parental organization.(3)Baby-centered questions: What happens to babies in the NICU, who takes care of them and their clinical trajectory.

These following quotes illustrate these themes:

“*I wish I had known how the unit worked before the delivery to prepare mentally (when I was on bed rest for a while before birth), also why all this happened to me and what I could have done to prevent prematurity*”(Mother of baby born at 26 weeks, now 18 months.corrected age);

“*I think people are hesitant to talk about prematurity when there is a risk thinking they will cause unnecessary stress, but it should be included in materials for a pregnant couple. Why did it happen to us and what we could have done? Then, once we are on bed rest, we could learn more practical things about the unit and how we can better prepare as parents*”(Mother of twins born at 28 weeks);

“*The hospitalization was really hard and there is no real way to prepare for that, but before birth, speaking about the emotions parents could have and that support exists in the intensive care could help. Feeling we are not alone in this experience. I still feel I could have prevented this, speaking about guilt would help*”(Mother of child born at 24 weeks, who is 3 years corrected); 

“*What to do as a father, I felt alone in all this and that I had to support my wife. There are things I could have done before birth to get ready and less lost in the unit. I would have also liked to know what a baby looked like, the machines around us, it was scary at first, so small and fragile and hooked on so many things that kept ringing*”(Father of baby born at 27 weeks, now at 20 months corrected.)

### 3.2. Pre-Pilot and “The COVID-19 Phase”

With the themes in the parental answers and following steps we used successfully for workshops in the NICU [8,9], we designed a checklist of the themes to be covered during the workshop (Appendix B) and made a PowerPoint presentation for parents that included pictures (Appendix B). These materials are also available in French (Appendix A).

The above-mentioned “Ombrelles” program was starting at the same time and included prenatal workshops. When we were about to initiate the program, in March 2020, the COVID-19 pandemic was beginning, meaning parents and clinicians were no longer allowed to meet in the same room for workshops. The only contact accepted between patients and clinicians were for “direct” clinical purposes only. At that time, pregnant women were only allowed to come to their prenatal visits or be hospitalized on the high-risk floors alone (fathers/the other parent or a significant person were only allowed to come for the delivery). All support groups in the hospital stopped. The workshops were therefore adapted to be held virtually and were opened to other participants (e.g., fathers, significant others). We developed several rules for virtual and hybrid workshops: (1) all cameras of virtual participants had to be on, throughout, (2) before every workshop, we reminded all participants that what was said was confidential and not to be repeated, (3) participants were free to speak or not. We provided iPads to the hospitalized women, if desired, and technical assistance to join the workshop (this was the beginning of virtual meetings when these skills were new for the majority of individuals: joining a meeting, muting the microphone, opening the camera, how to post questions in the chat, etc.).

Because of our previous experiences with workshops, we knew that parents participated more when it was called a workshop and not a support group or a meeting [8,9]. Holding the workshops weekly and always on the same day at the same time also improves attendance. The ideal time was found to be after lunch time, after medical clinical rounds, outside of routine laboratory hours (early morning or evening) and before nurses’ end of shift. The workshops took, and still take place, from 13 h 30–14 h 30 every Friday.

The pilot phase lasted longer than expected, 9 months, as we hoped that a rapid transition back to in-person workshops would occur. After each session we openly discussed what went well and how we could improve the workshop. This was the first time that a virtual workshop of this kind was piloted in our hospital. There were no instructions on how to lead virtual workshops in the literature and some were unsure whether virtual workshops should even occur. Initially, there were five workshop moderators with two moderating each workshop. Moderators optimized the moderator checklist and the visual presentation many times, following participant recommendations and their experience. When the content of the workshop had not been changed for six months, the moderators felt comfortable with the checklist, workshop format, and supporting documents, it was decided that one workshop moderator was sufficient. There are now 12 moderators capable of leading these workshops; each new moderator participated in about four workshops with another “experienced” moderator before feeling they could lead the workshop in an independent fashion.

Prospective parents had several recommendations and we developed additional materials. Because of COVID-19, it was impossible for them to visit the NICU antenatally nor to have access to a lactation consultant before birth (who were assigned at the bedside because of lack of clinicians). Therefore, we developed several videos: a virtual visit [13], how to initiate breastmilk expression after birth [14] and a video introducing the prenatal workshop [15]. They also wished to receive our NICU welcome package (made by NICU parents for new parents) antenatally [16]. Parents recommended that after the workshop we send them an e-mail with this information and links. The checklists were also finalized and additional topics that were always covered because of questions from parents were added, such as the existence of specific rooms for twins, and where parents could sleep at night (Appendix B; Appendix A).

### 3.3. Prospective Pilot Evaluation

During the pilot phase that started in January 2021, the perspectives of mothers from the first six (one month and a half) workshops were sought. Twenty-one mothers hospitalized in the high-risk pregnancy unit (average 32 years-old, pregnancy 29.3 weeks) responded. All agreed or strongly agreed (100%) that the workshop was useful, that it made them feel less lonely (95%), that exchanges with other women were beneficial (95%), that the workshop helped them prepare for the arrival of their baby (95%), and that it gave them a certain amount of control over their situation (89%).

When asked what they appreciated about the workshop through open-ended questions, prospective mothers invoked three themes: the information they learned, the interaction with other women in their situation, and the normalization of their emotions. The following quotes illustrate what participants appreciated:

“*All the information received was appreciated, it was very well planned. I liked the other participants. All the points raised about being parents in neonatology were so valid and I speak from experience, I had to go through the NICU in the past. Thank you so much by starting and mentioning guilt that pregnant women have.*”(32-year-old, in threatened preterm labor at 24-weeks);

“*I appreciated the emotional support and the answers to all my questions.*”(24-year-old women, 30-weeks pregnant with premature rupture of membranes)

“*I liked the fact that the moderator took the time to listen to us, reassure us and answered all our questions.*” (31-year-old, pregnant with twins at 29-weeks, her husband also participated);

“*The fact that it was interactive. We are not alone in our situation. We could ask questions to the professionals that were available for us.*”(38-year-old, 32 weeks pregnant with a diagnosis of preeclampsia, her husband also participated);

“*The main thing: to see we are not alone; many others are also going through these hardships.*” (23-year-old women in threatened preterm labor at 25 weeks);

“*It helped to understand what neonatology was, know what resources that will be available, then the video-visit of the unit that we watched after.*”(26-year-old, 32 weeks pregnant, starting induction for a diagnosis of severe intrauterine growth restriction);

“*I like the fact that they presented what is neonatology and showed us many images and spoke about each one together, described the machines and the process in general.*” (29-year-old, 28-weeks pregnant with rupture of membranes).

We also asked what participants wished to improve. All specified “nothing” except three who all reported they would rather have a workshop in person, but understood the challenges associated with COVID-19.

## 4. Discussion

Although prenatal classes are not new, when we planned this project, we could not find empirical practical information about how to develop and implement such meetings for parents facing a complicated pregnancy, birth and delivery. Moreover, there was no practical information about how to lead such workshops virtually. To our knowledge, we are the first to describe -in a practical fashion- the implementation of a prenatal workshop for parents who will likely have an infant admitted in NCIU. Several other support-group initiatives for parents of sick newborns have been described, but none antenatally [8,9,17,18,19,20,21,22]. Many parents -both during and after their NICU experience- reported wanting to meet other parents/prospective parents in their situation and discuss medical as well as “non-medical” themes, such as their parental roles, practical information and their emotions. Parents often grieve the loss of their desired “normal” pregnancy, delivery and healthy baby, and often believe that they could have prevented their preterm delivery [23]. Because NICU parents often have similar experiences, we suspect our checklist and themes could be used for developing prenatal workshops in other centers.

On the other hand, each hospital/medical center is unique and developing prenatal parent meetings requires an analysis of each unit to optimize parent satisfaction. Mainly because of COVID-19, the pre-pilot phase involved some trial and error. It was the first time such support groups happened virtually, and many clinicians felt uncomfortable. We were unsure whether speaking about intense emotions parents may feel, such as guilt, feelings of inadequacy, sadness, anger, etc. could be done in a virtual format. By being humble and transparent, and taking into consideration participants’ feedback, we were able to adjust rapidly and to further develop the structure and content of our meetings to better meet the needs of prospective parents. We had already developed workshops in the NICU and were prepared to face disappointments but remain motivated. For example, in our previous experiences, when we called the meetings “support groups” or placed them in the evening, we had few or no participants, which created serious disappointment [8,9].

Participants found the workshops useful, and particularly appreciated the fact that it decreased their isolation and made them feel like a community. Relationships between parents have been demonstrated to contribute to parent well-being and coping [24,25,26]. Many participants stayed in contact with each other, phoned each other daily (COVID-19 prevented them from seeing each other in person) on the prenatal ward, and later supported each other in the NICU. Participants also reported that the meetings normalized their feelings, prepared them for birth and improved their well-being. Prenatal workshops are only one of the many ways prospective parents can be supported.

Clinicians can help parents by providing counselling, social, psychological, and spiritual support, but participants in this study clearly reported major benefits of meeting others in their situation. When they shared stories with other parents who had experienced losses, trauma, infertility, pathological pregnancies, they saw they were not alone. Prospective parents can also get peer support from other “informal” sources, such as associations in the community or online support groups. We need to remember that these initiatives should not replace access to hospital support and individual support. We believe prenatal workshops should be offered to all prospective parents as a complement to professional support services.

## 5. Limitations

This study has several limitations. It was conducted in a single center and while the practical aspects of workshops in different hospitals may be different in other centers with different realities, the benefits of prenatal workshops probably transcend NICU realities and cultures. This investigation did not examine clinical impacts on parents and neonates, such as length of stay or psychological outcomes. We also only investigated the perspective of pregnant women admitted in a high-risk pregnancy unit, despite the fact that their significant others also attended these workshops. These meetings now take place in a hybrid fashion: many prospective parents come in person, for example women admitted in the high-risk pregnancy unit and prospective parents who will have a child in the NICU (for example with a congenital anomaly and are rarely admitted in the high-risk unit). Pregnant women are also invited to join online, such as women experiencing a pregnancy complication with home management through our Antenatal Home-Care unit. We will publish the perspective of prospective parents who have a child with congenital anomalies or diagnoses other than prematurity. During the groups, independent of diagnosis, parents speak about common themes reported in this study, such as parental guilt, future parental role, coping with many losses, adapting to their new reality, control and trust.

## 6. Conclusions

In conclusion, in this article we described practical information on how we developed prenatal workshops for parents who will likely have a newborn requiring NICU admission. We have herein included the moderator checklist we developed, as well as the relevant PowerPoint presentation (Appendix B; Appendix A); these are available for all those who wish to use it (please contact us if you need documents you may wish to adapt to your unit). Moreover, we demonstrated that it is feasible and beneficial to develop prenatal workshops for parents who will have an infant hospitalized in the NICU. These workshops are a unique and useful means to support prospective parents. Future investigations will inform us whether these initiatives have an impact on clinical and psychological outcomes.

## Data Availability

Not applicable.

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
