# Peer review of "Prenatal Workshops and Support Groups for Prospective Parents Whose Children Will Need Neonatal Care at Birth: A Feasibility and Pilot Study"

_children, 2023, doi:10.3390/children10091570_

Round 1
Reviewer 1 Report
i have read this paper with great interest and a background on clinical research in neonatology. I value the effort, but would like to put two aspects into perspective
first, this focusses mainly on preterm neonates, so how applicable or how different would this be for eg newborns with congenital malformations.
second, do we know if parents prefer 'groups', or 'individual interactions', or both ?
Author Response
Review of manuscript: Manuscript ID: children-2489210 entitled Prenatal workshops and support groups for prospective parents whose children will need neonatal care at birth: a feasibility and pilot study”
We want to thank the editor of Children and the reviewers of our paper for providing constructive comments and suggestions to improve our manuscript.
You will find below the list of each specific comments or recommendations made by reviewers and our answer to each. We have numbered the comments of reviewers (C1, C2, etc) followed by our answers (A1, A2) and descriptions of the changes in the manuscript. Our manuscript has been altered accordingly and enriched by these comments.
C1: I have read this paper with great interest and a background on clinical research in neonatology. I value the effort, but would like to put two aspects into perspective. first, this focusses mainly on preterm neonates, so how applicable or how different would this be for eg newborns with congenital malformations.
A1: We have acknowledged this in the limitation section, which now reads : “We also only investigated the perspective of pregnant women admitted in a high-risk pregnancy unit, despite the fact that their significant others also attended these workshops. These meetings now take place in a hybrid fashion: many prospective parents come in person, for example women admitted in the high-risk pregnancy unit and prospective parents who will have a child in the NICU (for example with a congenital anomaly and are rarely admitted in the high-risk unit). Pregnant women are also invited to join online, such as women experiencing a pregnancy complication with home management through our Antenatal Home-Care unit. We will publish the perspective of prospective parents who have a child with congenital anomalies or diagnoses other than prematurity. During the groups, independent of diagnosis, parents speak about common themes reported in this study, such as parental guilt, future parental role, coping with many losses, adapting to their new reality, control and trust”
C2:second, do we know if parents prefer 'groups', or 'individual interactions', or both ?
A2: As stated in the results, the group interaction was considered important to participants “All agreed or strongly agreed (100%) that the workshop was useful, that it made them feel less lonely (95%), that exchanges with other women were beneficial (95%),”. In the description of their perspectives, this exposed them to the normality of their emotions and they felt they were not alone in their situation. We did not ask participants whether they preferred groups or individual interactions. We suspect both are important and have different goals.
We have also optimized the discussion section as follows: “Participants found the workshops useful, and particularly appreciated the fact that it decreased their isolation and made them feel like a community. Relationships between parents have been demonstrated to contribute to parent well-being and coping (25-27) Many participants stayed in contact with each other, phoned each other daily (Covid prevented them from seeing each other in person) on the prenatal ward, and later supported each other in the NICU. Participants also reported that the meetings normalized their feelings, prepared them for birth and improved their well-being. Prenatal workshops are only one of the many ways prospective parents can be supported. Clinicians can help parents by providing counselling, social, psychological, and spiritual support, but participants in this study clearly reported major benefits of meeting others in their situation. When they shared stories with other parents who had experienced losses, trauma, infertility, pathological pregnancies, they saw they were not alone. Prospective parents can also get peer-support from other “informal” sources, such as associations in the community or online support groups. We need to remember that these initiatives should not replace access to hospital support and individual support. We believe prenatal workshops should be offered to all prospective parents as a complement to professional support services.”
Reviewer 2 Report
Dear authors,
Thank you for this very interesting and already well-written manuscript. I just have a few points that could be addressed to further improve it:
-) Introduction: I think this should be expanded a bit in terms of support group and existing literature.
-) Methods: Please also state the country your setting is in.
-) Results: This should be better structured I think, maybe with bullet points in the paragraphs?
-) Discussion: This, also, would benefit from more structuring, e.g., through subheadings. Also, a more in-depth comparison to already-existing literature on the topic (and similar domains where no literature is available) could enhance this section.
-) I would be interested in thoughts about your special setting circumstances - I assume they would be classified as "high-resource" settings. What about the need for support groups in low-resource settings?
-) Please mark the Conclusion as its own heading, and the Limitations as a subheading of the Discussion. This way, a reader will find them more easily.
Author Response
Review of manuscript, comments reviewer 2: Manuscript ID: children-2489210 entitled Prenatal workshops and support groups for prospective parents whose children will need neonatal care at birth: a feasibility and pilot study”
We want to thank the editor of Children and the reviewers of our paper for providing constructive comments and suggestions to improve our manuscript.
You will find below the list of each specific comments or recommendations made by reviewers and our answer to each. We have numbered the comments of reviewers (C1, C2, etc) followed by our answers (A1, A2) and descriptions of the changes in the manuscript. Our manuscript has been altered accordingly and enriched by these comments.
C1: Thank you for this very interesting and already well-written manuscript. I just have a few points that could be addressed to further improve it:
Introduction: I think this should be expanded a bit in terms of support group and existing literature.
A1). The literature on parent support groups in neonatology is scarce and we could find no article on such hospital-based education support groups for women on high-risk pregnancy units. We have reported on our peer-to-peer support groups in this article: “we recently developed workshops for parents in the NICU(8, 9). One of these workshops is called “being a parent in the NICU” and has been evaluated as beneficial by families.(9) The following were the main themes reported by parents when explaining why these meetings helped: decreasing isolation and becoming a community, getting practical information, promoting hope and resilience. Through sharing stories with other parents who also had experienced loss, sadness, and grief, parents trusted that it was possible to adapt and thrive. The meetings normalized parents' emotions, decreased negative emotions (eg, anger, sadness, guilt), empowered them in their parental role (and their added value in the clinical team caring for their child), and helped them communicate with loved ones and providers.”
We have expanded on the value of prenatal classes for “normal” pregnancies and this part of the instruction now reads as follows: Prospective parents in “normal” pregnancies have the opportunity to engage in prenatal classes as a group, but these classes or groups did not exist for parents expecting a sick baby requiring NICU care (nor for sick pregnant women with pathological pregnancies), nor did they address the potential needs of future “NICU parents”. There is also no literature on how to develop such prenatal workshops, nor the content that prospective parents may find useful. Prenatal classes and groups have educational aims (describing the pregnancy and delivery process) but also include a group experience where parents can engage with others in their situation.
C2) Methods: Please also state the country your setting is in.
A2). This has been added.
C3) Results: This should be better structured I think, maybe with bullet points in the paragraphs?
A3) These are now the bullet points
- Needs assessment
- Recommendations of parents who participated in postnatal workshops
- Recommendations from parents (the Parents’ Voice Study)
2.Prepilot and “the COVID phase”
3.Prospective pilot evaluation
C4) Discussion: This, also, would benefit from more structuring, e.g., through subheadings. Also, a more in-depth comparison to already-existing literature on the topic (and similar domains where no literature is available) could enhance this section.
A4) subheadings were added. There are (at the moment) no similar support groups for women hospitalized in the high-risk pregnancy unit. On the other hand, many support interventions exist for these women. This part of the discussion was optimized: “Prenatal workshops are only one of the many ways prospective parents can be supported. Clinicians can help parents by providing counselling, social, psychological, and spiritual support, but participants in this study clearly reported major benefits of meeting others in their situation. When they shared stories with other parents who had experienced losses, trauma, infertility, pathological pregnancies, they saw they were not alone. Prospective parents can also get peer-support from other “informal” sources, such as associations in the community or online support groups. We need to remember that these initiatives should not replace access to hospital support and individual support. We believe prenatal workshops should be offered to all prospective parents as a complement to professional support services.”
C5) I would be interested in thoughts about your special setting circumstances - I assume they would be classified as "high-resource" settings. What about the need for support groups in low-resource settings?
A5) Canada is a country where healthcare is universal. There are no private hospitals taking care of high-risk pregnancies as all NICUs are in the public sector. These hospitalized women are not from “high resource setting”, in fact there is an association with socio-economic status and high-risk pregnancies.
C6) Please mark the Conclusion as its own heading, and the Limitations as a subheading of the Discussion. This way, a reader will find them more easily.
A6) this has been done.
Reviewer 3 Report
This is an interesting and well-conducted study aiming to assess the benefit of support groups in the Neonatal Intensive Care Unit to parents. The usefulness of prenatal support groups for prospective parents who will have a newborn requiring admission to the NICU has not been investigated before. The study included a significant number of cases (295 parents) and several aspects, such as parental guilt, future parental role, coping with many losses, adaptation to the new reality, control and trust, information about the NICU, technology around the baby, common neonatal interventions, the NICU clinical team, and the role of parents in the team. Even more, initial findings were used to develop the workshop further, including a moderator checklist and a visual presentation. All women who answered the survey agreed that the workshop was useful, that it made them feel less lonely (95%), that exchanges with other women were beneficial (95%) and gave them a certain amount of control over their situation (89%). All answers to open-ended questions were positive. Conclusion is supported by the results: Prenatal educational/support workshops provide a unique and useful means to support future NICU parents. Limitations section is also included in the manuscript.
In general, literature review is adequate, methodology is valid and the results are clearly presented. I support the publication of this paper.
Author Response
There were no responses to provide for this reviewer who recommended publication of the paper.
Round 2
Reviewer 2 Report
No further comments - just one clarification: I meant high-resource settings as Canada = high-resource; various other parts of the world are not high-resource. Maybe you can add a sentence or two on this (= international applicability).
Author Response
Thanks very much for your comments. We did it.